materials science/nanotechnology

polydopamine, polytetrafluoroethylene, polyethyleneimine, surface charges, nanoparticles

**Authors for correspondence:**
Manjit Singh Grewal
e-mail: grewal.manjit.singh.d3@tohoku.ac.jp
Hiroshi Yabu
e-mail: hiroshi.yabu.d5@tohoku.ac.jp

This article has been edited by the Royal Society of Chemistry, including the commissioning, peer review process and editorial aspects up to the point of acceptance.

# Aqueous dispersion and tuning surface charges of polytetrafluoroethylene particles by bioinspired polydopamine–polyethyleneimine coating via one-step method

Manjit Singh Grewal[1], Hiroya Abe[2], Yasutaka Matsuo[3] and Hiroshi Yabu[1]

[1]WPI-Advanced Institute of Materials Research (WPI-AIMR), and [2]Frontier Research Institute for Interdisciplinary Sciences, Tohoku University, 2-1-1 Katahira, Sendai 980-8577, Japan
[3]Research Institute for Electronic Science (RIES), Hokkaido University, N21W10, Kita-Ku, Sapporo 001-0021, Japan

MSG, 0000-0002-5729-2724; HA, 0000-0002-8847-1581; YM, 0000-0002-5071-0284; HY, 0000-0002-1943-6790

We propose a surface modification of poorly dispersive polytetrafluoroethylene (PTFE) particles via bioinspired polydopamine–polyethyleneimine (PDA–PEI) which conferred PTFE particles a uniform dispersion in aqueous medium. With increasing dopamine concentration in the reaction solution, dispersity of PTFE particles improved and the surface charges of particles changed from negative to positive due to an increase of surface coverage of PDA–PEI layers. Simplicity of the method here outlines an attractive route for surface modification of inert surfaces useful for large-scale applications.

## 1. Introduction

Catechol groups, which are present in the adhesive proteins such as 3,4-dihydroxy-L-phenylalanine of marine mussels, show excellent adhesion properties in air and underwater for a wide variety of inert surfaces such as noble metals, inorganic metals, metal oxides, glass, ceramics, polymers and woods [1–6]. Mussel-inspired polydopamine (PDA), which is formed by the

oxidative polymerization of dopamine (DA) in aqueous alkaline media, has aroused great interest of researchers as a simple and powerful tool to functionalize chemically inert surfaces due to its versatility and great adhesive potential in many different materials [7–9]. The remarkable feature of PDA-based coating lies in their chemical structure, which can expose many functional groups such as amine, imine and catechol on the coated surface. These functional groups can support various chemical reactions with different functional groups and can serve as both the starting points for covalent modification [10–12] with desired molecules and the anchors for metal ions reduction. Sotoma and Harada reported PDA coatings to functionalize gold nanoparticles (NPs) [13] for ring-opening reactions. Fredi *et al.* has used PDA coatings for strong adhesives between paraffin microcapsules and an epoxy matrix [14] with low surface reactivity. The Zhu group [15] has reported PDA-coated carbon nanotubes conjugated to folic acid, through (1-ethyl-3-(3-dimethylaminopropyl)carbodiimide/ N-hydroxysuccinimide) activation coupling of amino groups of PDA and carboxylic groups of folic acids, for biomedical applications. In addition, PDA-based coatings have also been broadly used in endowing implant surface with excellent biocompatibility and bioactivity [16,17], in areas of science and engineering such as cell patterning [18], biosensing [19], stabilization of NPs [20], removal of heavy metals and organic pollutants [21], synthesis of photocatalyst [22] and fabrication of special wettable surfaces [23]. In order to realize the full potential of PDA coatings, strong research efforts are devoted to developing new biomimetic composite materials [24,25] for better adhesion and surface coatings or modifications owing to the growing needs and demands of automotive and aerospace engineering.

Polytetrafluoroethylene (PTFE), a synthetic hydrophobic fluoropolymer of tetrafluoroethylene, finds an exceptional position in a large number of successful engineering practical applications [26–29] due to its outstanding chemical and thermal resistance, low coefficient of friction, low surface energy, low dielectric constant, high resistivity, low diffusivity and potential biocompatibility. PTFE particles are commonly used as an additive for non-sticking and lubrication properties. To further improve its intrinsic properties for special purpose applications, PTFE particles are usually blended with other polymers or reinforced as a composite materials [30,31]. Surface modifications play an important role in improving or tuning the properties and overall performance of practically useful but chemically inert materials such as PTFE [5,6]. Techniques such as chemical etching [32], UV-laser exposure [33], electron and ion-beam irradiation [34], ozone treatment [6], plasma modifications [35] and radiation grafting [36] are tried to improve both its interfacial compatibility and dispersion in polymer matrixes. However, these approaches require complicated processes and expensive or specially made equipment. Therefore, it is a key problem to find a simple and effective way to modify PTFE. Additionally, it is also a challenging scope to disperse PTFE particles in an aqueous medium without using fluorinated surfactants. To address this issue, we previously reported non-ionic amphiphilic copolymers containing catechol groups in their side chains [12]. Moreover, some of the previous works have suggested to coat inert substrates with PDA layer through oxidative self-polymerization of DA [34,37] using electrodeposition technique or in organic solvents.

In the light of aforementioned potential of PDA, we fabricated PDA–polyethyleneimine (PDA–PEI) coating on PTFE particles based on spontaneous oxidative polymerization and successfully dispersed PTFE particles in aqueous medium. Keeping the amount of PEI fixed, varying amounts of DA were conjugated via Michael addition or Schiff base reaction. Then the effects of PDA–PEI coating on surface charge of PTFE composites were evaluated in detail. The results showed that PDA–PEI coatings were successfully fabricated on the surface of PTFE particles, due to which uniform dispersion of PTFE particles was possible in aqueous medium. Further, we noticed change in surface charge of PTFE@PDA–PEI particles based on compositional control of PDA as shown in figure 1. The current approach constitutes a straightforward methodology requiring minimal engineering to incur hydrophilicity and produce surface-charged PTFE particles based on PDA–PEI coating. We envisage that surface-modified PTFE particles can have great potential in a wide range of applications across the chemical, biological, medical and materials sciences, as well as in applied science engineering and the technology fields.

# 2. Experimental set-up

## 2.1. Materials

Dopamine hydrochloride was purchased from Sigma Aldrich, St Louis, USA PEI (average mol. weight 600) and 28% ammonia solution were purchased from Fujifilm Wako Pure Chemical Corporation,

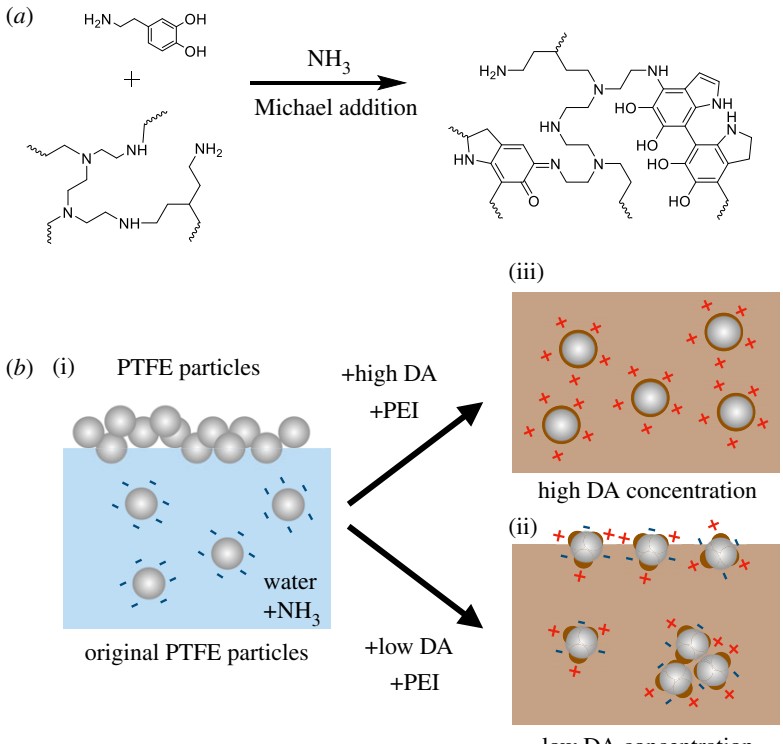

**Figure 1.** (a) Reaction scheme for PDA–PEI composite; (b) schematic illustration of PTFE particle dispersion without (i)/with (ii), (iii) PDA–PEI coating.

Osaka, Japan. PTFE NPs with average diameter of 100 nm were purchased from Okuno Chemicals Industries Co., Ltd, Osaka, Japan. All the chemicals were used as received.

## 2.2. Methods

Morphology of PDA–PEI-coated PTFE composites (PTFE@PDA–PEI) and pristine PTFE particles were observed by using scanning electron microscopy (SEM) (JEOL JSM-6700F) using an 1–2 kV electron beam. Fourier transform infrared (FT-IR) transmission spectra were recorded on a JASCO FT-IR-6100 between 4000 and 600 cm$^{-1}$ using a diamond attenuated total reflection accessory. X-ray photoelectron spectroscopy (XPS, KRATOS AXIS-Ulta DLD, Shimazu, Japan) was used for determining an elemental composition, chemical state and electronic state of the elements within the composite materials. Thermogravimetric analysis (TG-DTA) was performed on RIGAKU Thermo Plus EvoII TG-DTA8210 at a heating rate of 10°C min$^{-1}$ under N$_2$ atmosphere. Dynamic light scattering (DLS) sizing measurements and surface charge determination experiments were performed by employing a Zetasizer Nano ZS (Malvern ZEN3600).

## 2.3. Preparation of aqueous dispersive polydopamine–polyetheyleneiminine-coated polytetrafluoroethylene (PTFE@PDA–PEI) particles

Firstly, 100 ml of ammonia solution with pH above 10 containing 200 mg (2 g l$^{-1}$) of DA hydrochloride, 1 g (10 g l$^{-1}$) of PEI was prepared. PTFE particles (100 mg) were added to the above-mentioned alkaline solution and homogenized for 30 min at room temperature. A chemical route between DA and PEI is shown in figure 1. The same procedure was used to prepare variations by varying the concentration of DA as 400, 600, 800 mg, 1, 2, 3 g keeping all the constituents same.

## 3. Results and discussion

Table 1 summarizes dispersion of PTFE particles in the presence of constituting materials, ammonia water, DA, PEI, the absence of any of these leads to non-dispersion of PTFE particles. Corresponding

(*a*)

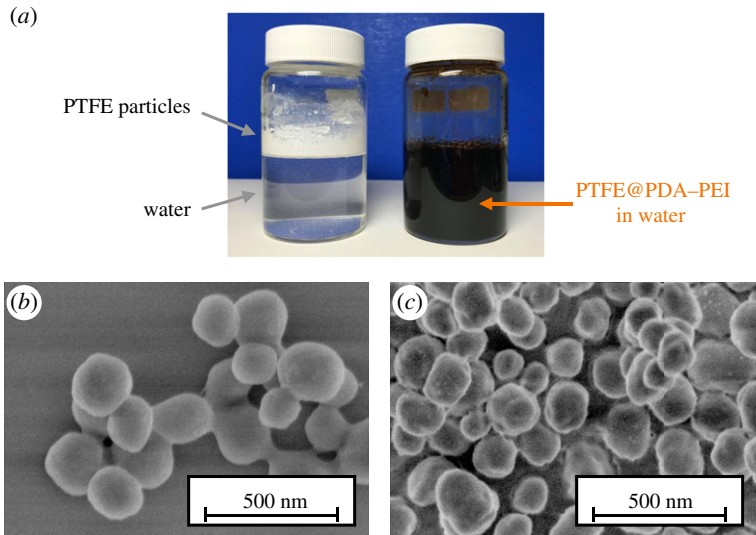

PTFE particles

water

PTFE@PDA–PEI
in water

(*b*)

500 nm

(*c*)

500 nm

**Figure 2.** A photograph of pristine PTFE particles in water and PTFE@PDA–PEI particles in an aqueous medium (*a*). SEM images of pristine PTFE particles (*b*) and PTFE@PDA–PEI particles (*c*).

**Table 1.** Examination of PTFE particle dispersion in the presence of ammonia water, dopamine, polyethyleneimine. (●: includes, –: excludes).

| sample | 1 | 2 | 3 | 4 |
|---|---|---|---|---|
| PTFE particle aq. (1 g l$^{-1}$) | ● | ● | ● | ● |
| NH$_3$ aq. (pH 10.5) | ● | ● | ● | ● |
| DA (2 g l$^{-1}$) | – | ● | – | ● |
| PEI (10 g l$^{-1}$) | – | – | ● | ● |

figures are shown in electronic supplementary material, figure S1. Direct dispersion of pristine PTFE particles (1 g l$^{-1}$) in an aqueous medium, even though DA was mixed, was difficult even with the assistance of prolonged sonication (sample 1–3), while PDA–PEI-coated PTFE particles, PTFE@PDA–PEI (sample 4), showed uniform dispersion in aqueous medium at alkaline pH. Dispersion was evaluated with compositional changes of PDA. With low concentration of PDA, little floated particles were observed (electronic supplementary material, figure S2). Unlike pristine PTFE particles, which showed very large agglomerates on the surface of water, PTFE@PDA–PEI particles dispersed very well in water, forming a homogenous dispersion system (see electronic supplementary material, figure S1*b*). This reveals that the surface of PTFE@PDA–PEI became noticeably hydrophilic. The chemical route for PDA–PEI composite and sketched illustration are shown in figure 1.

Figure 2*a* shows the digital photograph of PTFE and PTFE@PDA–PEI particles in aqueous medium. Scanning electron microscopy images of PTFE and PTFE@PDA–PEI are shown in figure 2*b*,*c*. As corroborated from the SEM images, spherical morphology of pristine PTFE particles is clearly distinguished from the crumpled or granule-like surface of PTFE@PDA–PEI due to uniform surface coating by PDA–PEI. FT-IR spectra of pristine PTFE and PTFE@PDA–PEI are shown in figure 3*a*. The characteristic symmetric and asymmetric stretching of CF$_2$ was observed at 1148 (iv) and 1203 cm$^{-1}$ (v), respectively. Further, the absorption peaks at 639 (iii), 553 (ii) and 504 (i) cm$^{-1}$ are due to CF$_2$ wagging, bending and rocking whose intensity changes simultaneously with the crystallinity degree of PTFE. The simultaneous decrease of intensity of above-mentioned peaks and the appearance of broad bands around 3353 cm$^{-1}$ (viii) due to –OH and amine groups of PDA–PEI, and around 1590 cm$^{-1}$ (vi) due to the aromatic rings of PDA supporting the wrapping of PTFE by PDA–PEI. Also, $\sigma_{C-H}$ from PEI mainchain appeared around 2900 cm$^{-1}$ (vii) confirming composite formation with PEI.

X-ray photoelectron spectroscopy was used to detect the chemical elements of modified PTFE. The results of the XPS spectra of PTFE with the surface modification of DA with 0.10 and 3 g l$^{-1}$ were tested and are shown in figure 3*b*–*f*. The original PTFE particles only had their unique elements:

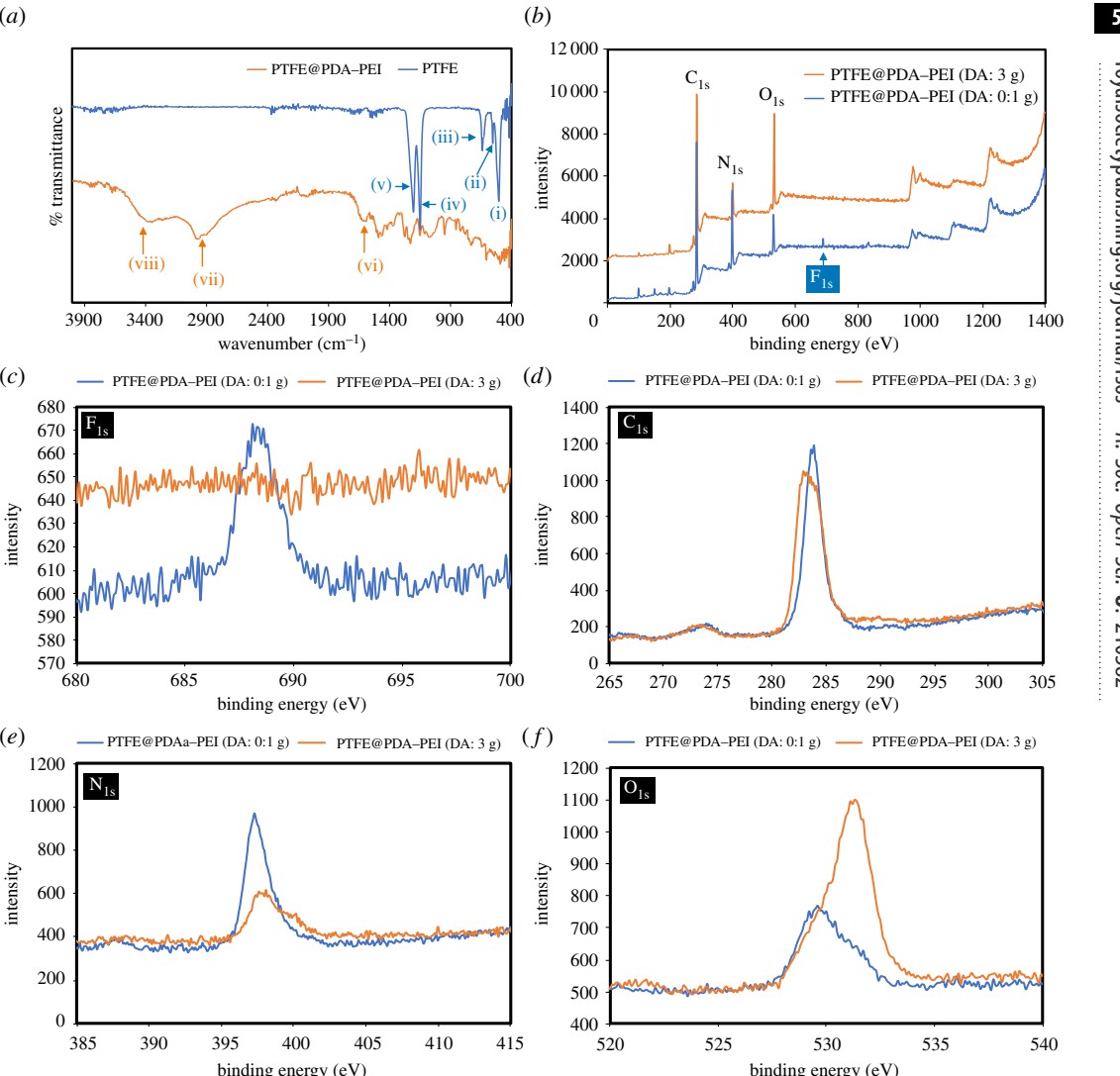

**Figure 3.** FT-IR spectra of pristine PTFE and PTFE@PDA–PEI (*a*), XPS spectra of PTFE@PDA–PEI with different concentrations of dopamine (*b*) and F$_{1s}$ (*c*), C$_{1s}$ (*d*), N$_{1s}$ (*e*) and O$_{1s}$ (*f*) spectra.

(*c* and *f*). It is worth noting that a significant O-element absorption peak appeared at about 532 eV in the spectra of PTFE@PDA–PEIs. The F-element is the endemic element of the PTFE. Additionally, PDA–PEI contains hydroxyl groups and other oxygen containing groups that resulted in the increase of oxygen content in the PTFE@PDA–PEI. The XPS data show clearly increase of O-element signal and decrease of F-element signal, indicating the formation of PDA/PEI shells on the surface of PTFE. Further, broadening of the N-element absorption peak at about 400 eV shows increased contribution from PDA–PEI coatings. These results proved that PTFE was successfully wrapped by a layer of PDA–PEI. Surface-modified PTFE particles were further evaluated for surface charge. It was observed that the PTFE@PDA–PEI particles underwent a change in their surface charge from negative to positive with an increase in concentration of DA in the PDA–PEI composite. This phenomenon could happen presumably due to increased synergistic effect of PDA–PEI and hydrogen bonding (H-F) between amine and hydroxy groups. The results are summarized in table 2 and in figure 4. Analytical results of particle size determination by DLS sizing measurements of PTFE@PDA–PEI particles with different concentration of DA and pristine PTFE particles are provided in electronic supplementary material, figure S3. It was observed that under the low concentration of DA, bimodalities appear due to the presence of uncoated particles but as the concentration of DA increased, the distribution became unimodal. A comparison was made with pristine PTFE particles in water in electronic supplementary material, figure S3 along with sketched representation in figure 1*b*.

The thermal properties of PTFE and PTFE@PDA–PEI were evaluated by TG-DTA analysis (electronic supplementary material, figure S4) in a nitrogen atmosphere at a heating rate of 10°C min$^{-1}$. The thermal

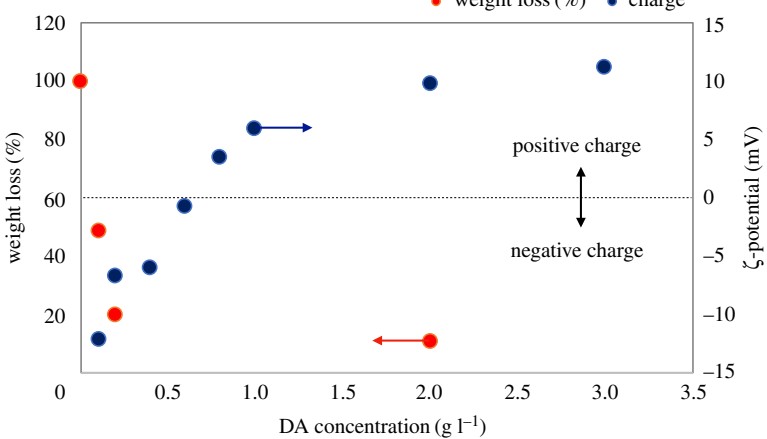

**Figure 4.** Effect of concentration of dopamine on surface charge of PTFE@PDA–PEI (blue circles) and weight loss@500°C in TG-DTA (red circles).

**Table 2.** Summary of surface charge (mV) and DLS measurements with change in concentration of dopamine in PTFE@PDA–PEI keeping all other constituents (PEI: 1 ml, ammonia water: 1 ml; PTFE: 100 mg) fixed.

| sample | DA (g l$^{-1}$) | $\zeta$-potential (mV) | $_{ave}D_h$ (nm) |
|---|---|---|---|
| 1 | 0.1 | −12.17 | 960 |
| 2 | 0.2 | −6.94 | 1767 |
| 3 | 0.4 | −5.99 | 2331 |
| 4 | 0.6 | −0.71 | 549 |
| 5 | 0.8 | 3.54 | 1073 |
| 6 | 1.0 | 5.96 | 648 |
| 7 | 2.0 | 9.78 | 209 |
| 8 | 3.0 | 11.14 | 479 |

degradation of pristine PTFE starts at a very high temperature of above 450°C leaving almost negligible residue at the end of degradation. The residual 5% weight loss ($T_5$) temperature for PTFE is as high as 510°C and 10% weight loss ($T_{10}$) temperature is at 524°C. Where $T_5$ and $T_{10}$ for PTFE@PDA–PEI are 146 and 215°C respectively, leaving high residue at the end of degradation due to wrapping of PDA–PEI shells, unlike pristine PTFE. The results of the thermogravimetric analysis are in good agreement with the changes in surface charge based on compositional variation of DA. Primarily, the assembly of the PDA–PEI thin layer is formed by the spontaneous oxidative polymerization of DA onto PTFE particles. Under a DA concentration of 1 g l$^{-1}$, the thickness of the PDA–PEI template increases and the PTFE particles are coated with multiple layers, through sequential adsorption of polymers from the solution, which changed the surface charge of PTFE particles from negative to positive. Over DA concentration of 1.0 g l$^{-1}$, the tendency of increase in surface charge and residual percentage became moderate owing to the uniform coverage of PTFE surface by PDA–PEI layers. The residual percentage goes down with DA and the zeta potential goes up with DA. The summarized results are shown in figure 4.

In conclusion, in the projected study we reported dispersion and surface charge modification of PTFE particles in aqueous medium by mussel-inspired PDA co-deposited with PEI in alkaline medium. In conventional PDA coatings, PDA coatings were either based on sacrificial polymer layers or using organic solvents or electrodepositions. The growth rate for PDA films in most cases was very slow and/or film thickness was typically less than 100 nm. The addition of PEI into the alkaline solution of PDA increased the thickness via cross-linking reactions between PDA and PEI. The rate of PDA film formation was much faster. PEI can cross-link PDA via the Michael addition reaction and the Schiff base reaction. In alkaline solution DA, which is a catechol containing amine compound, oxidizes to

quinone form and forms the PDA–PEI coating on PTFE. Based on the intrinsic characteristic of the PTFE material, they could be combined with PDA through strong intermolecular forces, such as coordinate bond, H-F and van der Waals forces. This work proves that the surface modification of PTFE particles by PDA–PEI coating is a powerful technique to realize one-step fabrication of functional coatings with improved surface hydrophilicity, chemical stability and morphological uniformity as evidenced by FT-IR and XPS spectra. Further, the surface charge modification of PTFE was evaluated by conjugating varying amounts of DA in PDA–PEI composites keeping PEI as constant. Our results put forward new evidence that PDA–PEI coating exerts direct influence on surface charge modification of inert surfaces such as PTFE particles used as templates in the present work. As a bioinspired polymer, PDA–PEI coatings can be used as a potential functional bioactive coating for diverse templates or inert surfaces and modification of surface charge. These results may be extended further as a powerful route for surface and charge modification of various inert surfaces that can be exploited for different applications. Such charged surfaces can be used for different industrial, biomedical or nanotechnological applications such as metallization or mineralization.

Data accessibility. In the manuscript, we present new data and the instructions how to reproduce the experiments are fully explained in the main text of the manuscript and electronic supplementary material. Electronic supplementary material is related to the photographs of PTFE@PDA–PEI particles before and after dispersion in influence of different constituents, supporting data of DLS and thermogravimetry measurements, SEM images of PTFE@PDA–PEI particles with DA concentration: 0.1, 0.2, 0.4 and 0.6 g l$^{-1}$, and graphical abstract.

The data are provided in electronic supplementary material [38].

Authors' contributions. M.S.G. and H.A. synthesized PDA–PEI composites to disperse PTFE particles. Y.M. measured XPS to analyse chemical composition. M.S.G. and H.Y. wrote the whole part of the manuscript. Whole work was conducted by H.Y.

Competing interests. There are no conflicts to declare.

Funding. H.Y. thanks KAKENHI, JSPS and MEXT (grant nos. 20H04625, 19KK0357, 18H05482) for financial support. H.A. thanks KAKENHI (grant no. 19K15598) for financial support.

Acknowledgements. Dr M. S. Grewal and Prof. Hiroshi Yabu thank technical staff Ms. Mayumi Sasaki and Ms. Shizuka Miyamoto of Yabu laboratory at WPI-AIMR, Katahira Campus, Tohoku University for DLS and SEM measurements. We are further grateful to Common Equipment Facilities of WPI-AIMR, Katahira Campus, Tohoku University to allow us to use TG-DTA instrument for thermogravimetric analysis.

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
