## [Peer Review File · Royal Society Open Science]

Review History

RSOS-210582.R0 (Original submission)

Review form: Reviewer 1

Is the manuscript scientifically sound in its present form?

Yes

Are the interpretations and conclusions justified by the results?

Yes

Is the language acceptable?

Yes

Do you have any ethical concerns with this paper?

No

Have you any concerns about statistical analyses in this paper?

No

Recommendation?

Accept with minor revision (please list in comments)

Comments to the Author(s)

This paper describes the surface modification of superhydrophobic PTFE particles with polydopamine-polyethylenimine (PDA-PEI). Although there are many reports on the modification of the PTFE surface by conventional PDA coating, this study shows the modification of the PTFE particle surface in a single step in an aqueous system. The procedure is simple and may extend the range of use of PTFE particles. Therefore, the reviewer is recommended to publish this article with appropriate modifications.

1. What is the difference between a conventional PDA coating and a PDA-PEI coating?

Please mention the interaction between the PDA-PEI thin film and the surface of the PTFE particles. The large contact area of the coated film has been reported to induce strong adhesion of the film to the substrate (see Soft Matter 2011, 7, 1856, Macromol. Rapid Commun., 2013, 34, 1220). In addition to the chemical structure, physical effects can also affect the coating. Does the molecular weight of PEI affect the adhesive properties?

2. The stability of the PDA-PEI coating is important. Is PDA-PEI detachment observed when reanalyzed after 1 week or 1 month?

Review form: Reviewer 2

Is the manuscript scientifically sound in its present form?

Yes

Are the interpretations and conclusions justified by the results?

Yes

Is the language acceptable?

Yes

Do you have any ethical concerns with this paper?

No

Have you any concerns about statistical analyses in this paper?

No

Recommendation?

Major revision is needed (please make suggestions in comments)

Comments to the Author(s)

This paper described the surface modification of PTFE particles that can disperse in water uniformly. Precisely tuned concentration of DA and PEI successfully modified the surface charge between positive and negative of the PTFE particles. The reaction mechanism is very straightforward, and thus this kind of surface engineering onto polymers can be applied into the various practical applications. The reviewer accept this manuscript of RSOS-210582 after major revision as noted.

1. For Figure 1, the reviewer suggests that addition of water contact angle measurements of sample after spin coating or drop casting the (i), (ii), and (iii) onto glass substrate for reader's better understanding in surface charge density change part.

2. How much of the thickness of the wrapped PDA-PEI shells on the PTFE?
3. For Figure 2, the size distribution diagram of PTFE and PTFE@PDA-PEI particles analyzed and calculated directly by SEM images should be useful to compare the dynamic light scattering spectra. This reviewer feels that the size of particles are not matched well between SEM and DLS.
4. For Figure 2, is there any difference between PTFE and PTFE@PDA-PEI when conduct 1D wide angle X-ray diffraction (WAXD) measurements after lyophilization?
5. For Figures 3 and 4, they are hard to get information because of low resolution. Please fully revise them, particularly x- and y-axis.
6. The author stated that PTFE@PDA-PEI can be used for industrial and biomedical application. Could you please explain more in detail because the changing charge density of conventional materials is very effective to pave the new way for development of antifouling finish or filter membrane.
7. For Page 2 right column, space is missing for "2g" and "3g".
8. For Page 4 left column, it should be typo error for "(T10)".

Decision letter (RSOS-210582.R0)

Dear Professor Yabu:

Title: Aqueous Dispersion and Tuning Surface Charges of Polytetrafluoroethylene Particles by Single-Step Bioinspired Polydopamine-Polyethyleneimine Coating
Manuscript ID: RSOS-210582

The editor assigned to your manuscript has now received comments from reviewers. We would like you to revise your paper in accordance with the referee and Subject Editor suggestions which can be found below (not including confidential reports to the Editor). Please note this decision does not guarantee eventual acceptance.

Please submit your revised paper before 18-Jun-2021. Please note that the revision deadline will expire at 00.00am on this date. If we do not hear from you within this time then it will be assumed that the paper has been withdrawn. In exceptional circumstances, extensions may be possible if agreed with the Editorial Office in advance. We do not allow multiple rounds of revision so we urge you to make every effort to fully address all of the comments at this stage. If deemed necessary by the Editors, your manuscript will be sent back to one or more of the original reviewers for assessment. If the original reviewers are not available we may invite new reviewers.

On behalf of the Subject Editor Professor Anthony Stace and the Associate Editor Professor Chaohua Cui.

RSC Associate Editor:
Comments to the Author:
(There are no comments.)

RSC Subject Editor:
Comments to the Author:
(There are no comments.)

Reviewers' Comments to Author:
Reviewer: 1

Comments to the Author(s)

This paper describes the surface modification of superhydrophobic PTFE particles with polydopamine-polyethylenimine (PDA-PEI). Although there are many reports on the modification of the PTFE surface by conventional PDA coating, this study shows the modification of the PTFE particle surface in a single step in an aqueous system. The procedure is simple and may extend the range of use of PTFE particles. Therefore, the reviewer is recommended to publish this article with appropriate modifications.

1. What is the difference between a conventional PDA coating and a PDA-PEI coating? Please mention the interaction between the PDA-PEI thin film and the surface of the PTFE particles. The

large contact area of the coated film has been reported to induce strong adhesion of the film to the substrate (see *Soft Matter* 2011, 7, 1856, *Macromol. Rapid Commun.*, 2013, 34, 1220). In addition to the chemical structure, physical effects can also affect the coating. Does the molecular weight of PEI affect the adhesive properties?

2. The stability of the PDA-PEI coating is important. Is PDA-PEI detachment observed when reanalyzed after 1 week or 1 month?

Reviewer: 2

Comments to the Author(s)

This paper described the surface modification of PTFE particles that can disperse in water uniformly. Precisely tuned concentration of DA and PEI successfully modified the surface charge between positive and negative of the PTFE particles. The reaction mechanism is very straightforward, and thus this kind of surface engineering onto polymers can be applied into the various practical applications. The reviewer accept this manuscript of RSOS-210582 after major revision as noted.

1. For Figure 1, the reviewer suggests that addition of water contact angle measurements of sample after spin coating or drop casting the (i), (ii), and (iii) onto glass substrate for reader's better understanding in surface charge density change part.
2. How much of the thickness of the wrapped PDA-PEI shells on the PTFE?
3. For Figure 2, the size distribution diagram of PTFE and PTFE@PDA-PEI particles analyzed and calculated directly by SEM images should be useful to compare the dynamic light scattering spectra. This reviewer feels that the size of particles are not matched well between SEM and DLS.
4. For Figure 2, is there any difference between PTFE and PTFE@PDA-PEI when conduct 1D wide angle X-ray diffraction (WAXD) measurements after lyophilization?
5. For Figures 3 and 4, they are hard to get information because of low resolution. Please fully revise them, particularly x- and y-axis.
6. The author stated that PTFE@PDA-PEI can be used for industrial and biomedical application. Could you please explain more in detail because the changing charge density of conventional materials is very effective to pave the new way for development of antifouling finish or filter membrane.
7. For Page 2 right column, space is missing for "2g" and "3g".
8. For Page 4 left column, it should be typo error for "(T10)".

Author's Response to Decision Letter for (RSOS-210582.R0)

See Appendix A.

RSOS-210582.R1 (Revision)

Review form: Reviewer 1

Is the manuscript scientifically sound in its present form?

Yes

Are the interpretations and conclusions justified by the results?

Yes

Is the language acceptable?

Yes

Do you have any ethical concerns with this paper?

No

Have you any concerns about statistical analyses in this paper?

No

Recommendation?

Accept as is

Comments to the Author(s)

The revised manuscript is clearly written on the points suggested by reviewers. Thus, this manuscript is acceptable for publication in Royal Society Open Science.

Review form: Reviewer 2

Is the manuscript scientifically sound in its present form?

Yes

Are the interpretations and conclusions justified by the results?

Yes

Is the language acceptable?

Yes

Do you have any ethical concerns with this paper?

No

Have you any concerns about statistical analyses in this paper?

No

Recommendation?

Accept as is

Comments to the Author(s)

The revised version and author's response is acceptable.

Decision letter (RSOS-210582.R1)

Dear Professor Yabu:

Title: Aqueous Dispersion and Tuning Surface Charges of Polytetrafluoroethylene Particles by Single-Step Bioinspired Polydopamine-Polyethyleneimine Coating
Manuscript ID: RSOS-210582.R1

It is a pleasure to accept your manuscript in its current form for publication in Royal Society Open Science. The chemistry content of Royal Society Open Science is published in collaboration with the Royal Society of Chemistry.

On behalf of the Subject Editor Professor Anthony Stace and the Associate Editor Professor Chaohua Cui.

RSC Associate Editor:
Comments to the Author:
(There are no comments.)

RSC Subject Editor:
Comments to the Author:
(There are no comments.)

Reviewer(s)' Comments to Author:

Reviewer: 1

Comments to the Author(s)

The revised manuscript is clearly written on the points suggested by reviewers. Thus, this manuscript is acceptable for publication in Royal Society Open Science.

Reviewer: 2

Comments to the Author(s)

The revised version and author's response is acceptable.

Appendix A

Tohoku University
Sendai, Miyagi 980-8577, Japan

June 5, 2021

Professor Dr Laura Smith
Journal Editor
Royal Society Open Science

Prof. Hiroshi Yabu
AIMR, Tohoku University
2-1-1, Katahira, Aoba, Sendai, 980-8577, JAPAN
TEL/FAX: +81-22-217-6341/6342
E-mail: hiroshi.yabu.d5@tohoku.ac.jp

Dear Prof. Dr Laura Smith,

We once again greatly appreciate your time dealing with the manuscript entitled “*Aqueous Dispersion and Tuning Surface Charges of Polytetrafluoroethylene Particles by Single-Step Bioinspired Polydopamine-Polyethyleneimine Coating*”. The constructive advice of the peer reviewers and the editorial office have substantially improved our paper.

As suggested by the editorial office, all the changes in the manuscript have been marked with red color and highlighted, and the 2 versions (revised manuscript and clean versions) are submitted to the online system.

We hope the revised manuscript will better suit the journal, Royal Society Open Science but are happy to consider further revisions. Please let me know of your decision at your earliest convenience. We thank you for your continued interest in our research.

With best regards
On behalf of the authors
Yours sincerely,
Hiroshi Yabu

Response to Referees

Reviewers' Comments to Author:

Reviewer: 1

Comments to the Author(s)

This paper describes the surface modification of superhydrophobic PTFE particles with polydopamine-polyethylenimine (PDA-PEI). Although there are many reports on the modification of the PTFE surface by conventional PDA coating, this study shows the modification of the PTFE particle surface in a single step in an aqueous system. The procedure is simple and may extend the range of use of PTFE particles. Therefore, the reviewer is recommended to publish this article with appropriate modifications.

We very much appreciate the reviewer's constructive comments and suggestions on our manuscript. We have extensively revised the manuscript to your response. Our responses (in red fonts) follow the reviewer's comments (in *italics*).

1. What is the difference between a conventional PDA coating and a PDA-PEI coating? Please mention the interaction between the PDA-PEI thin film and the surface of the PTFE particles. The large contact area of the coated film has been reported to induce strong adhesion of the film to the substrate (see Soft Matter 2011, 7, 1856, Macromol. Rapid Commun., 2013, 34, 1220). In addition to the chemical structure, physical effects can also affect the coating. Does the molecular weight of PEI affect the adhesive properties?

Authors' response: In conventional PDA coatings, PDA coatings were either based on sacrificial polymer layers or using organic solvents or electrodepositions. The growth rate for polydopamine films in most cases was very slow and/or film thickness were typically less than 100 nm. In the present study, the addition of polyethyleneimine (PEI) into alkaline solution of PDA increased the thickness via cross-linking reactions between PDA and PEI. The rate of polydopamine film formation was much faster. PEI can cross-link polydopamine via the Michael addition reaction and the Schiff base reaction. In an alkaline solution dopamine, which is a catechol containing amine compound, oxidizes to quinone form and forms the polydopamine (PDA)-PEI coating on PTFE. Based on the intrinsic characteristic of the PTFE material, they could be combined with PDA through strong intermolecular forces, such as coordinate bond, hydrogen bond (H-F) and Van der Waals forces. Molecular weight and concentration of PEI did affect the adhesive properties. In the present case, concentration of PEI below 10g/L led to non-dispersion or non-uniform coating.

2. The stability of the PDA-PEI coating is important. Is PDA-PEI detachment observed when reanalyzed after 1 week or 1 month?

Authors' response: The reviewer's indication is quite reasonable. Cross-linked PDA-PEI coatings were quite stable and hence detachment was not observed.

We greatly appreciate reviewer's time in reviewing the script. Thank you so much.

Response to Referees

Reviewers' Comments to Author:

Reviewer: 2

Comments to the Author(s)

This paper described the surface modification of PTFE particles that can disperse in water uniformly. Precisely tuned concentration of DA and PEI successfully modified the surface charge between positive and negative of the PTFE particles. The reaction mechanism is very straightforward, and thus this kind of surface engineering onto polymers can be applied into the various practical applications. The reviewer accept this manuscript of RSOS-210582 after major revision as noted.

We would like to thank the reviewer for the careful and thorough reading of this manuscript. The comments are encouraging, and the reviewer appears to share our judgement about the present study. The reviewer's comments are in *italics*, and the revised sentences in the manuscript were coloured in red.

1. For Figure 1, the reviewer suggests that addition of water contact angle measurements of sample after spin coating or drop casting the (i), (ii), and (iii) onto glass substrate for reader's better understanding in surface charge density change part.

Authors' response: The authors appreciate the reviewer for the above suggestions. The contact angles along with standard deviations (CA \pm SD) for pristine PTFE (i), PTFE@PDA-PEI with DA as 0.2 g/L (ii) and PTFE@PDA-PEI with DA as 2 g/L (iii) were 135.5 ± 3.6 , 51.3 ± 2.2 and 22.3 ± 4.6 , respectively.

2. How much of the thickness of the wrapped PDA-PEI shells on the PTFE?

Authors' response: The thickness of wrapped PDA-PEI shells are in the range of 100 nm to 500 um from SEM images.

3. For Figure 2, the size distribution diagram of PTFE and PTFE@PDA-PEI particles analyzed and calculated directly by SEM images should be useful to compare the dynamic light scattering spectra. This reviewer feels that the size of particles are not matched well between SEM and DLS.

Authors' response: The reviewer's concern on the standard deviation on the peaks, in this case, is quite understandable. The DLS graphs show the result as an intensity distribution. In the same sample, due to possible presence of particles of different size. As a result, larger particles scatter much more light than small particles, the intensity of scattering of a particle is proportional to the sixth power of its diameter (from Rayleigh's approximation). The values obtained from dls measurements with change in concentration of dopamine as shown in the Table 2 are the overall range with uncertainty considered. The reason for such high tolerance

is due to the Zetasizer system that determines the size by first measuring the Brownian motion of particles in a sample using Dynamic Light Scattering (DLS) and then interpreting a size from this correlation function (based on Stokes-Einstein Equation). The Zetasizer software uses algorithms to extract the decay rates for a number of size classes (arising from the amount of PDA-PEI coating on PTFE particles) to produce a size distribution. Whereas SEM images shows the clear PTFE particles coated with PDA-PEI shells with thickness ranged from 100 nm to 500 nm. The authors prepared samples multiple times in each with DA concentration.

4. *For Figure 2, is there any difference between PTFE and PTFE@PDA-PEI when conduct 1D wide angle X-ray diffraction (WAXD) measurements after lyophilization?*

Authors' response: In the present study, An X-ray photoelectron spectroscopy (XPS, KRATOS AXIS-Ulta DLD, Shimazu, Japan) was used for determining an elemental composition, chemical state and electronic state of the elements within the composite materials. WAXD measurements of samples after freeze drying or lyophilization was not carried out but should present considerable 2θ differences or intensity.

5. *For Figures 3 and 4, they are hard to get information because of low resolution. Please fully revise them, particularly x- and y-axis.*

Authors' response: On reviewer's indication, the resolutions for Figures 3 and 4 were improved.

6. *The author stated that PTFE@PDA-PEI can be used for industrial and biomedical application. Could you please explain more in detail because the changing charge density of conventional materials is very effective to pave the new way for development of antifouling finish or filter membrane.*

Authors' response: Polytetrafluoroethylene (PTFE) is one of the most important candidate of engineering polymers owing to its high chemical inertness, high temperature resistance, low temperature resistance, excellent heat endurance and strong mechanical strength. However strong hydrophobicity limits its wide applications. By coating PDA-PEI onto PTFE surface, its hydrophilicity and compatibility with other materials can be improved which makes it to be a good candidate for the applications in the fields of water treatment, chemical industry, textile, electronics, medical treatment, military, aerospace and so on. Further, specific functional groups can be designed and incorporated into the surface of PTFE@PDA-PEI particles to allow subsequent surface functionalization, such as enzyme and protein immobilization via covalent bonding.

7. For Page 2 right column, space is missing for "2g" and "3g".

Authors' response: Spacings were corrected in 2 g and 3 g.

8. For Page 4 left column, it should be typo error for "(T10)".

Authors' response: T_{10} was corrected.

We sincerely thank the reviewer again for their valuable comments, which were of great help in revising the manuscript.